# Association of University Students’ COVID-19 Vaccination Intention with Behaviors toward Protection and Perceptions Regarding the Pandemic

**DOI:** 10.3390/medicina58101438

**Published:** 2022-10-12

**Authors:** Chrysoula Dafogianni, Freideriki Eleni Kourti, Ioannis Koutelekos, Afroditi Zartaloudi, Evangelos Dousis, Areti Stavropoulou, Nikoletta Margari, Georgia Toulia, Despoina Pappa, Polyxeni Mangoulia, Eftychia Ferentinou, Anna Giga, Georgia Gerogianni

**Affiliations:** 1Department of Nursing, University of West Attica, 12243 Athens, Greece; 2School of Social Sciences, Hellenic Open University, 26335 Patra, Greece; 3School of Medicine, National and Kapodistrian University of Athens, 10679 Athens, Greece; 4Department of Nursing Specialties and Education, Evangelismos General Hospital, 10676 Athens, Greece

**Keywords:** COVID-19 pandemic, behavioral changes, protection, perceptions, personal risk, vaccination intention

## Abstract

*Background and Objectives:* The COVID-19 pandemic is a serious global health problem. Vaccination is suggested to be one of the most efficacious precautionary measures, in combination with other protective behaviors. The purpose of this study was to explore the association of students’ intention to get vaccinated about COVID-19 with protection behaviors and perceptions about the pandemic. *Material and Methods:* The study sample included 1920 university students who electronically completed two validated questionnaires anonymously and voluntarily from December 2020 to January 2021. *Results:* Results of the multiple linear regression analysis showed that as the perceived general risk was getting lower, the number of protective behaviors significantly diminished (*p* < 0.001). Additionally, respondents who believed that they had minor or no personal risk had undertaken significantly fewer preventing behaviors in comparison with participants who thought they had major personal risk (*p* = 0.006). However, the experience of respondents with people having COVID-19 had statistically significant association with undertaking more preventing behaviors (*p* = 0.004). Lower general perception of risk had statistically significant association with lower determination to obtain the vaccines of COVID-19 (*p* < 0.001). Personally knowing someone who had the coronavirus and undertaking more behavioral changes due to the coronavirus situation were significantly related to greater determination to obtain the vaccines of COVID-19 (*p* = 0.005 and *p* < 0.001, respectively). *Conclusions:* The results of this study can provide universities with the appropriate information about the improvement of COVID-19 vaccination strategies.

## 1. Introduction

COVID-19 has been identified as “the most severe global health challenge since the Spanish Flu one century ago” [1], as nearly 5.94 million people have died from the coronavirus until 31 December 2021 [2]. Vaccination is suggested to be one of the most efficacious precautionary measures, in combination with social distancing, isolation, appropriate sanitation rules, and use of masks in order to limit the spread of the coronavirus [3,4].

Universities are considered high-risk areas for COVID-19 outbreaks, as coronavirus can easily prevail in the crowded environment of campuses [5]. As a result, university students have an increased risk to transmit the virus to other people [6]. However, a large proportion of students have seemed to hesitate to be vaccinated, according to several studies [7,8,9]. Hesitation to vaccination is regarded as “the delay in acceptance or vaccination refusal despite its availability” and is included in top 10 threats in world health, according to the World Health Organization [10].

Barello et al. [11] concluded that 14% of Italian university students exhibited low intention of being vaccinated. Additionally, Faase and Newby [12] concluded that younger age (18–29 years) was related to lower association with behaviors about health protection. Similar studies including university students have reported lower percentages of seasonal flu vaccination and lower knowledge regarding vaccines [13,14].

Students’ response toward vaccination is influenced by the way they perceive, evaluate, and understand the potential risk [1]. Risk perception has been suggested to be a determining factor that influences individuals’ responses to the pandemic [15]. The risks of contracting the coronavirus have been underestimated by young adults, who consider themselves invulnerable to the effects of the disease [16]. Given the fact that students are young, are healthy, have fewer comorbidities, and usually have mild symptoms after their infection, they seem to differently understand disease risks compared with older adults [17]. Due to their belief in their invulnerability, they may engage in health behaviors that can increase their risk of becoming a source of infection for other people and their family members [18]. Several studies concluded that over 80% of young people complied with precautionary measures mainly motivated by their desire to be socially responsible and protect other people rather than personal perceived risk [19,20,21,22]. Nevertheless, Barello et al. [11] have also noted that “students are a good target for educational campaigns as they are still in their training period and are open to changing their habits.” Understanding students’ perceptions regarding the pandemic and how these affect the adoption of protective behaviors would help healthcare professionals implement more specific and targeted interventions to promote vaccination campaigns and restrict the transmission of COVID-19.

The purpose of this study was to explore the association of students’ intention to get vaccinated about COVID-19 with protection behaviors and perceptions about the pandemic.

## 2. Material and Methods

### 2.1. Data Collection

The present study was electronically carried out with university students from December 2020 to January 2021. The questionnaires were converted into an online file via an electronic questionnaire creation platform on the official site of the University of West Attica and then posted in an electronic communication environment. Participation was voluntary and anonymity was assured. All the respondents gave an informed consent. Before the collection of data, we received approval from the Research Committee of the University of West Attica.

Two validated questionnaires were used to collect data, which are presented as follows: 

A validated questionnaire indicated by Sherman et al. [23] was used to investigate attitudes and beliefs about COVID-19-preventing behavior and COVID-19 vaccination.

A validated questionnaire indicated by Seale et al. [24] was applied in this research study including demographic characteristics of participants, possible compliance with virus protection, beliefs toward quarantine, and perceived efficiency of infection control measures. The questionnaires were adjusted to the purpose of this study.

The above research questionnaires were licensed with authorized permission for usage from the scientific authors. Cultural weighting of the above questionnaires for adjustment to the Greek language was carried out. A reverse translation was made to the questionnaires from English to Greek and vice versa by two independent translators following the appropriate instructions for translating the questionnaire into the Greek language.

The inclusion criteria for participants were to be students attending university schools and be able to speak, read, and write in Greek. Exclusion criteria were inadequate language skills and age under 17 years old.

### 2.2. Statistical Analysis

Mean (standard deviation) or median (interquantile range) is used for describing quantitative variables, whereas absolute and relative frequencies were used for describing qualitative variables. Finding independent factors associated with the number of behavioral changes due to the pandemic and students’ intention to get vaccinated against COVID-19 was carried out via multiple linear regression analysis. Students’ demographics and their perceptions on the COVID-19 pandemic were used as independent variables for both dependent variables, whereas in the case where students’ intention to get vaccinated against COVID-19 was the dependent variable, the number of behavioral changes that students undertook was also included in the analysis as an independent variable. Due to numerous missing values, the item “As far as you know, is there currently a widely available vaccination to protect against coronavirus?” was not included in the regression analysis. From the linear regression models, adjusted regression coefficients (β), beta coefficients, and their confidence intervals were emerged. Statistical significance was set at *p* < 0.01 due to a large number of predictors in the model. Assumptions of the analyses were checked. *p*-values are two-tailed, and analyses were undergone via SPSS statistical software (IBM Corp., Armonk, NY, USA; version 22.0).

## 3. Results

The sample included 1920 respondents (73.1% women) with a mean age of 20.8 years (SD = 4.1 years). Most of the students were Greek (94.8%) and unmarried (97.0%). Furthermore, 20.6% of the students belonged to an extremely clinically vulnerable group, and 66.1% had an extremely clinically vulnerable person in their family (Table 1).

Concerning the students’ perception, almost half of the sample (49.3%) thought that COVID-19 is a significant danger to Greek individuals, and 36.7% thought that this risk on a personal level was minor. Only 1.9% of the participants referred that they had the coronavirus at some point, and 66.8% personally knew someone who had the coronavirus. In addition, 31.5% of the participants were informed about available vaccines against the coronavirus (Table 2).

Concerning the students’ preventing behaviors, more common practices reported by the respondents were the avoidance of crowded areas (88.2%), reduction of the use of public transportation (87.2%), and frequently washing their hands (81.2%). Most of the respondents (98.3%) undertook at least one preventing behavior. The number of preventing behaviors ranged from 0 to 8, and the mean was 5.6 (SD = 1.7). The majority of the respondents (40.9%) used to undertake seven preventing behaviors (Table 3).

## 4. Factors Independently Associated with the Number of Students’ Behavioral Changes

The results of the multiple linear regression analysis showed that as the perceived general risk was getting lower, the number of preventing behaviors significantly diminished (*p* < 0.001). Concerning the perceived personal danger, the results showed that the respondents who believed that they had minor or no personal danger had undertaken significantly fewer preventing behaviors compared with the participants who believed they had major personal danger (*p* = 0.006). However, the experience of the respondents with people having COVID-19 had statistically significant association with undertaking more preventing behaviors (*p* = 0.004) (Table 4). 

The intention of participants to get vaccinated against COVID-19 ranged from 0 (strongly disagree) to 10 (strongly agree), with a mean value of 5.3 (SD = 3.5) and a median of 5 (IQR: 2–9) (Figure 1).

The results of the multiple linear regression analysis showed that older students had significantly lower intention to get vaccinated (*p* = 0.002). Lower perceived danger due to the coronavirus was significantly related to lower intention to obtain the vaccines of COVID-19 (*p* < 0.001). Personally knowing someone who had the coronavirus and undertaking more behavioral changes due to the coronavirus situation were significantly related to the increased intention to obtain the vaccines of COVID-19 (*p* = 0.005 and *p* < 0.001, respectively) (Table 5).

## 5. Discussion

In this study, the majority of the participants used to undertake seven preventing behaviors due to the COVID-19 pandemic such as avoidance of crowded areas, reduction of the use of public transportation, and frequently washing their hands, findings that are congruent with similar studies [25,26].

This study showed that as the perceived general danger was getting lower, the number of preventing behaviors significantly decreased. Additionally, the participants who believed that they had minor or no personal danger due to COVID-19 used to follow significantly fewer preventing behaviors compared with the respondents who believed they had a major personal danger. It is worth noting that the way people perceive COVID-19 is significantly associated with their worries and preventing behaviors [27]. In a similar study, it was found that less than 45.7% of nursing students had adequate preventive behaviors toward COVID-19 [28], while Shahwan et al. [29] found that more than half of the students were misunderstanding the most frequent symptoms of COVID-19, such as nausea, vomiting, and diarrhea, and had little knowledge about the transmission of COVID-19. However, participants of an increased age had better COVID-19 preventive behavior than young participants possibly due to their high perceived susceptibility [28].

This study also showed that the experience of participants with people having COVID-19 had statistically significant association with undertaking more preventing behaviors. This can be viewed in the context of the provision of healthcare since it has been found that people who are involved in health-related professions have an increased intention to get vaccinated against COVID-19 as they have a high level of knowledge about this field, and they are aware of the severity of vaccination to improve the pandemic [30].

In this study, it was found that students with more increased age had significantly lower intention to get vaccinated. The most frequent reasons for students’ low intention to get vaccinated were their necessity for further research, unrevealed side effects of vaccines, and their opinion that vaccination was inefficient to limit the danger of infection [31,32]. Similarly, Shahwan et al. [29] indicated that students had low confidence in COVID-19 vaccination and preferred vaccines to be domestically manufactured instead of imported.

In another study, a negative approach of college students toward COVID-19 vaccination was found even though they were highly exposed to risks. This can be possibly attributed to their fear of side effects and long-term complications of vaccines, as well as their low level of self-efficacy in protecting themselves [33]. However, higher perceived seriousness of the coronavirus was related to more concern about it, leading to increased vaccination intention and actual vaccine uptake [34]. 

The results of the present study indicated that lower perception of the danger due to COVID-19 was significantly related to lower intention to obtain the vaccines. This may be due to inadequate information since Abdel-Aziz et al. [32] found in their study that 66.7% of medical students did not obtain adequate information about COVID-19 vaccines. In a similar study, 68.9% of the participants were hesitant to get vaccinated due to low information about it [26].

This study also showed that students who used to undertake more preventive behaviors due to the coronavirus situation were significantly related to greater intention to get vaccinated. It has been found that the preventive behavior of nursing students is related to the knowledge they receive about COVID-19 transmission and prevention in the university where they study [35]. Thus, the appropriate education about pandemic risk perception can help students comply with COVID-19 preventive practices [28]. Moreover, websites of government and international organizations, social media, and published scientific articles can provide adequate information about COVID-19 prevention measures and vaccination to university students [32]. Finally, healthcare providers can play a vital role in influencing university students about vaccine recruitment [36]. 

The effectiveness of the vaccine has been answered by the scientific community in various phases of vaccination implementation. The vaccine against COVID-19 has been proved to be effective in strengthening the immunity of the person who receives it in combination with the implementation of preventive measures. Regarding safety, the vaccine went through several phases before its initial application to the population. In general, the vaccines appeared to be safe in their application except for special cases of citizens who had individual special features. The motivation to apply vaccination to citizens derives from the incentive of scientists to participate in the vaccination process and the strategy of governments to use incentives to raise awareness and encourage citizens to carry out vaccination. The cost of implementation is covered by the health policy of the governments of each country [37,38].

### Limitations of the Study

This study has some limitations. Although the online survey may have limited the representativeness of the study sample, a large sample size was included in an attempt to address this problem. Moreover, in the present study, a large portion of the participants were women.

## 6. Conclusions

The results of the present study concern the first period of the pandemic. The present study showed that as the perceived general danger was getting lower, the number of preventing behaviors significantly decreased. The participants who believed that they had minor or no personal danger due to COVID-19 used to follow significantly fewer preventing behaviors compared with the respondents who believed they had a major personal danger. Lower perception of the danger due to COVID-19 was significantly related to lower intention to obtain the vaccines. However, the experience of the respondents with people having COVID-19 had statistically significant association with undertaking more preventing behaviors and greater intention to get vaccinated, which emphasizes the importance of social influence and other people’s suffering in the adoption of health-related behaviors. Social environment can affect students’ attitudes. Consequently, health communicative strategies implemented by government and international organizations, social media, and healthcare providers in the context of public health campaigns should emphasize on the protection of family, friends, and community from the suffering caused by COVID-19. Additionally, websites of government and international organizations, social media, published scientific articles, and healthcare providers should provide adequate and clear information about COVID-19 prevention measures and vaccination safety and effectiveness to university students. Transparency, clarity, and consistency of communication from the aforementioned organizations regarding COVID-19 pandemic dimensions are of great importance in order to ensure university students’ trust to encourage their vaccine uptake and protective behaviors.

## Figures and Tables

**Figure 1 medicina-58-01438-f001:**
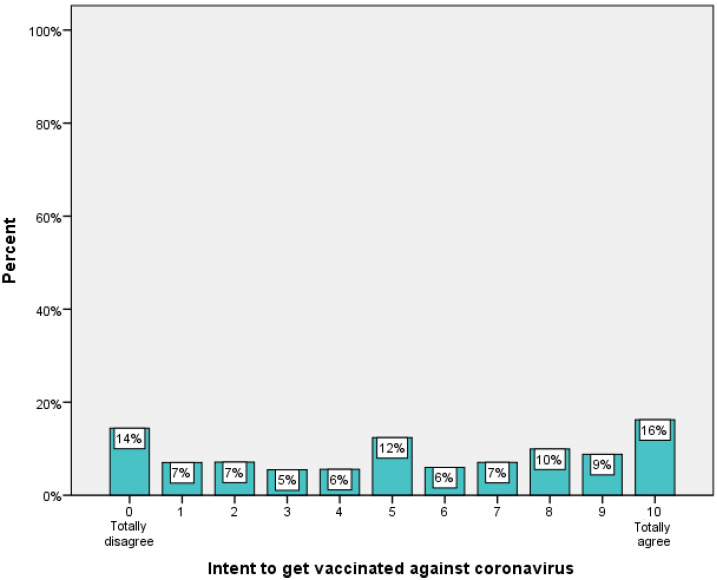
Students’ intention to get vaccinated against coronavirus.

**Table 1 medicina-58-01438-t001:** Characteristics of participants.

		*n* (%)
Sex	Men	517 (26.9)
	Women	1403 (73.1)
Age, mean (SD)		20.8 (4.1)
Nationality	Greek	1821 (94.8)
	Other	87 (4.6)
	Prefer not to say	12 (0.6)
Unmarried		1846 (97.0)
Children		24 (1.3)
Religion	Christian Orthodox	1375 (72.1)
	No religion	218 (11.4)
	Other	43 (3.5)
	Do not know	1 (0.1)
	Prefer not to say	245 (12.8)
Residence	Athens or Attica	1463 (76.8)
	Out of Athens	418 (21.9)
	Out of Greece	25 (1.3)
Flu vaccination last winter	No	1545 (80.5)
	Yes	276 (14.4)
	Do not know	90 (4.7)
	Prefer not to say	9 (0.5)
Belong in extremely clinically vulnerable group of people		395 (20.6)
Having an extremely clinically vulnerable person in family		1271 (66.1)

**Table 2 medicina-58-01438-t002:** Respondents’ perceptions about coronavirus.

Item	Level	*n* (%)
To what extent do you believe that COVID-19 is a danger to Greek individuals?	Major danger	349 (18.2)
	Significant danger	948 (49.3)
	Moderate danger	481 (25)
	Minor danger	120 (6.2)
	No danger at all	12 (0.7)
	Do not know	11 (0.6)
To what extent do you believe COVID-19 is a danger to you personally?	Major danger	101 (5.3)
	Significant danger	333 (17.3)
	Moderate danger	603 (31.4)
	Minor danger	706 (36.7)
	No risk at all	159 (8.3)
	Do not know	20 (1)
Do you think that you have had, or currently have, COVID-19?	Definitely	37 (1.9)
	Probably	161 (8.4)
	Probably not	709 (36.9)
	Definitely not	766 (39.9)
	Do not know	244 (12.7)
	Prefer not to say	4 (0.2)
Do you personally know anyone (excluding yourself) who has had COVID-19?	No	634 (33)
	Yes	1284 (66.8)
	Prefer not to say	4 (0.2)
As far as you know, are there any available vaccines against COVID-19?	No	770 (40.6)
	Yes	598 (31.5)
	Do not know	467 (24.6)
	Prefer not to say	61 (3.3)

**Table 3 medicina-58-01438-t003:** Preventing behaviors toward COVID-19.

	*n* (%)
Washed my hands with soap and water more frequently than usual	1561 (81.2)
Used alcoholic gel for hands more than usual	1520 (79.2)
I increased the amount I clean or disinfect things I might touch, like doorknobs	1331 (69.3)
Kept away from crowded areas	1693 (88.2)
Reduced the amount I use public transportation	1676 (87.2)
Cancelled or postponed a social event, such as meeting friends, eating out, or going to a sports event	1411 (73.5)
Reduced the amount I go into shops	1501 (78.2)
Kept one or more of my children out of school or preschool ^1^	12 (50.0)
Undertake ≥ 1 preventing behaviors due to COVID-19	1890 (98.3)
Number of preventing behaviors due to COVID-19, mean (SD)	5.6 (1.7)

^1^ Referred in students with children (*n* = 24).

**Table 4 medicina-58-01438-t004:** Multiple linear regression analysis results having the number of behavioral changes due to COVID-19 pandemic as a dependent variable and students’ characteristics and their perceptions for COVID-19 pandemic as independent variables.

	Unstandardized	Standardized		
	Coefficient β	Coefficient b	99% CI for β	*p*
Sex (reference: Men)	0.09	0.02	–0.16; 0.34	0.341
Age	–0.01	–0.02	–0.04; 0.02	0.513
Nationality (reference: Greek)	–0.50	–0.06	–1.08; 0.08	0.027
Unmarried (yes vs. no)	0.08	0.01	–0.76; 0.91	0.806
Children (yes vs. no)	0.58	0.04	–0.70; 1.85	0.243
Religion (reference: no religion)				
Christian Orthodox	–0.22	–0.05	–0.53; 0.10	0.078
Other	–0.31	–0.04	–0.92; 0.30	0.195
Residence (reference: Athens or Attica)				
Out of Athens	0.07	0.02	–0.20; 0.33	0.504
Out of Greece	–0.71	–0.05	–1.75; 0.33	0.078
Flu vaccination last winter (yes vs. no)	0.11	0.02	–0.20; 0.42	0.365
Belong in extremely clinically vulnerable group of people (yes vs. no)	–0.08	–0.02	–0.37; 0.20	0.461
Having an extremely clinically vulnerable person in family (yes vs. no)	0.02	0.01	–0.22; 0.26	0.819
To what extent do you think COVID-19 is a danger to Greek individuals? (Reference: major danger)				
No danger at all or minor danger	–2.23	–0.34	–2.76; –1.69	<0.001 **
Moderate danger	–0.97	–0.25	–1.36; –0.58	<0.001 **
Significant danger	–0.34	–0.10	–0.66; –0.02	0.006 *
To what extent do you believe COVID-19 is a danger to you? (Reference: major danger)				
No danger at all or minor danger	–0.60	–0.17	–1.16; –0.03	0.006 *
Moderate danger	–0.03	–0.01	–0.57; 0.51	0.901
Significant danger	–0.02	0.00	–0.56; 0.53	0.942
Do you think you have had, or currently have, COVID-19? (Reference: definitely/probably)				
Probably not	0.16	0.05	–0.19; 0.51	0.242
Definitely not	0.07	0.02	–0.28; 0.43	0.583
Do you know anyone (excluding yourself) having COVID-19? (Yes vs. no)	0.26	0.07	0.03; 0.50	0.004 *

* *p* < 0.01; ** *p* < 0.001.

**Table 5 medicina-58-01438-t005:** Multiple linear regression analysis results having students’ intention to get vaccinated against COVID-19 as dependent variable and their characteristics, their perceptions for the pandemic, and number of behavioral changes that they undertook due to pandemic as independent variables.

	Unstandardized	Standardized		
	Coefficient β	Coefficient b	99% CI for β	*p*
Sex (reference: men)	–0.33	–0.04	–0.83; 0.17	0.091
Age	–0.07	–0.09	–0.13; –0.01	0.002 *
Nationality (reference: Greek)	0.39	0.02	–0.77; 1.56	0.384
Unmarried (yes vs. no)	–0.71	–0.04	–2.38; 0.96	0.270
Children (yes vs. no)	–0.59	–0.02	–3.14; 1.96	0.552
Religion (reference: no religion)				
Christian Orthodox	–1.37	–0.15	–2.00; –0.74	<0.001 **
Other	–0.91	–0.05	–2.14; 0.31	0.055
Residence (reference: Athens or Attica)				
Out of Athens	0.10	0.01	–0.43; 0.62	0.643
Out of Greece	–0.22	–0.01	–2.30; 1.85	0.780
Flu vaccination last winter (yes vs. no)	1.03	0.11	0.41; 1.65	<0.001 **
Belong in extremely clinically vulnerable group of people (yes vs. no)	–0.16	–0.02	–0.73; 0.41	0.471
Having an extremely clinically vulnerable person in family (yes vs. no)	0.19	0.03	–0.29; 0.66	0.313
To what extent do you believe COVID-19 is a danger to Greek individuals? (Reference: major danger)				
No danger at all or minor danger	–4.64	–0.35	–5.76; –3.52	<0.001 **
Moderate danger	–2.70	–0.33	–3.49; –1.90	<0.001 **
Significant danger	–1.25	–0.18	–1.89; –0.61	<0.001 **
To what extent do you believe COVID-19 is a danger to you? (Reference: major danger)				
No risk at all or minor danger	–0.51	–0.07	–1.63; 0.62	0.244
Moderate danger	–0.35	–0.05	–1.43; 0.73	0.401
Significant danger	–0.16	–0.02	–1.25; 0.93	0.707
Do you think you have had, or currently have, COVID-19? (Reference: definitely/probably)				
Probably not	0.18	0.02	–0.53; 0.88	0.516
Definitely not	–0.29	–0.04	–0.99; 0.41	0.287
Do you know anyone (excluding yourself) having COVID-19? (Yes vs. no)	0.52	0.07	0.05; 0.99	0.005 *
Number of behavioral changes due to coronavirus situation	0.26	0.13	0.11; 0.40	<0.001 **

* *p* < 0.01; ** *p* < 0.001.

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
