# Peer review of "Association of University Students’ COVID-19 Vaccination Intention with Behaviors toward Protection and Perceptions Regarding the Pandemic"

_medicina, 2022, doi:10.3390/medicina58101438_

Round 1
Reviewer 1 Report
Although the end of the COVID-19 pandemic is in sight, however, it still needs to be vaccinated and boosted for the virus. Dafogianni et al. reported an interesting manuscript that may provide universities with the appropriate information about the improvement of COVID-19 vaccination strategies. It can be acceptance after having some minor changes:
1. How about considering vaccine efficacy and safety, incentive for vaccination and cost was performed from the study? I think it should be included further.
2. The conclusion of the study is not clear, and it should be taken based on the current pandemic.
3. There are some similar approaches that have got published recently, please make sure to include all and discuss them well. Examples: doi:10.1001/jamanetworkopen.2022.12681; https://doi.org/10.1371/journal.pone.0255447
4. Proof of the questionnaire should be provided as a Supplemental
5. Since Table 4 is not well presented, it would be better to convert the Table to a figure using R+
Author Response
Please see the attachement.

Reviewer 2 Report
The authors have comprehensively reviewed the association between intention to get vaccine and behaviors toward protection and perception. The manuscript is generally easily readable.
Few points.
1. The sentence in the abstract, 'However, meeting people with COVID-19 was statistically significantly associated with undertaking more preventing behaviors (p=0.004).', needs improvement. I think the word 'experience' should be included in the sentence. It took quite a lot of time to figure out the meaning of 'meeting people with COVID-19'. At a glance, it reads as if the people who will meet people with COVID-19 undertakes more preventing measures or behaviors.
2. line 41: I'm not sure the this is a intended spelling 'virous'. Consider using virus, instead.
3. line 43: 'Despite of this' is somewhat odd. Please consider improving the conjunction.
4. lines 75–76: 'Electronic communication environment' should be detailed in specific. Was it on the official site of the Universities? or was it on Facebook or Twitter? This may be a significant part of the study method.
5. line 133: I find that there are a large portion of women in the survey (73.1%). The authors should check the percentage of female students enrolled in major universities in Greece. The survey might have been somewhat biased.
6. line 184, line 191: 'it was found' is oddly repeated. Please consider improving the repetition.
7. The authors should consider limitations of their study, including the one mentioned above (item 5).
English is editing is need.
Round 2
Reviewer 1 Report
It now can be acceptance!